# Supervised Pre-training for Unsupervised Product-Patent Image Retrieval

## Abstract

Detecting infringing products is essential for protecting intellectual property rights and is often implemented as a product-patent retrieval task. Manual infringement detection is extremely time-consuming, and artificial intelligence plays an increasingly important role. However, most existing methods rely on natural language-based retrieval due to the domain discrepancies between patent images and product images. Due to the lack of sufficient annotated data, this work aims to address the aforementioned issues in an unsupervised setting by answering the following two questions: 1) How can we align the domain gap between patent images and product images using existing technologies? 2) How can we build a powerful backbone to jointly extract the features of patent and product images? Initially, we construct a dataset for patent-product image retrieval, which includes product-patent pairs and unlabeled data. To address the first question, we systematically evaluate three unsupervised approaches to mitigate the domain gap between patent and product images. The results demonstrate that jointly mapping patent and product images to a new feature space is effective. To answer the second question, we propose a novel supervised pre-training paradigm to achieve domain-aligned feature extraction for product and patent edge images. Extensive experiments using various backbones and training pipelines demonstrate the superiority of our supervised pre-training method. The dataset and code of this paper will be made publicly available upon acceptance.

## 1 Introduction

In recent years, the rapid development of artificial intelligence (AI) has revolutionized various fields, including natural language processing (Dong et al., 2022; Achiam et al., 2023), computer vision (Kirillov et al., 2023), medical AI (Celard et al., 2023; Gong et al., 2023; Huang et al., 2024), and intellectual property protection (Krestel et al., 2021; Shomee et al., 2024). As an essential component of intellectual property rights, patents have garnered increasing attention, and numerous AI technologies in natural language processing (Kang et al., 2020; Higuchi & Yanai, 2023) and computer vision (Kravets et al., 2017; Higuchi et al., 2023) have been applied to them. For example, during the patent application process, examiners can utilize AI to retrieve textual content (e.g., abstracts, main bodies) (Shomee et al., 2024) and images of patents to ensure that newly applied patents do not infringe upon previously authorized ones—an effort often defined as patent classification retrieval (Kang et al., 2020).

After a patent is granted, an important task is to ensure that new products emerging in the market do not infringe upon existing patents in the database—a topic that has received relatively less discussion (Krestel et al., 2021). In this context, most patent-related applications are based on natural language processing technologies (Lu et al., 2020; Yücesoy Kahraman et al., 2023; Li et al., 2023; Bekamiri et al., 2024; Suzgun et al., 2024). Companies responsible for intellectual property protection, for instance, determine whether a product infringes on a patent by comparing product descriptions with patent descriptions (Radford et al., 2019). However, according to our research, methods that retrieve patent images using product images are seldom mentioned, primarily due to domain discrepancies: patent images are usually represented by line drawings and sketches, whereas product images are predominantly colorful RGB images captured by cameras. This domain difference hinders more efficient infringing product retrieval.

To address this issue, we pose two critical questions that need to be considered in the patent-product retrieval task:

1. **How can we alleviate the domain discrepancies between patents and products?** This is a representation problem; we need to determine a representation method that can mitigate the domain differences between patents and products since the retrieval is performed in an unsupervised manner.

2. **Why does supervised pre-training outperform unsupervised pre-training in the unsupervised patent-product retrieval task?** This is a learning problem; we aim to train a supervised feature encoder that can better learn high-dimensional features conducive to retrieval.

## 1.1 A1: Alleviating Domain Discrepancies through Feature Mapping

Mapping product and patent images into a new feature space helps alleviate domain discrepancies. There are three strategies to align the image features of products and patents:

1. **Converting abstract patent line drawings into product images:** The challenge here is that patent line drawings are binarized sketches (Shomee et al., 2024), whereas product images contain rich color information. Current generative models may incorrectly colorize patent line drawings—for example, assigning inaccurate colors to specific components—which leads to discrepancies between the colorized images and actual product images (as shown in Fig. 1).

2. **Transforming product images into abstract line drawings:** This approach faces the limitations of existing algorithms. Patent line drawings are often composed of continuous lines, but current edge extraction algorithms struggle to produce continuous edges in the extracted results, introducing domain discrepancies.

3. **Mapping both product and patent images into a common feature space:** By utilizing an edge extraction algorithm to simultaneously extract edge features from both patent and product images, this method effectively mitigates the domain discrepancies highlighted in the second strategy. We adopt this strategy, as it offers the advantage of reducing the domain gap by mapping the product and patent images into the same feature space. Further analysis and experiments are presented in Fig. 1 and Fig. 3.

## 1.2 A2: Advantages of Supervised Pre-training over Unsupervised Pre-training

In the context of unsupervised patent-product retrieval, a significant challenge is the scarcity of large-scale paired products and patent images for training. To address this, we opt to use alternative proxy tasks, such as classification and segmentation, to train a backbone network for edge feature extraction (Zhou et al., 2024). Unsupervised pre-training methods (e.g., DINO (Caron et al., 2021), iBOT (Zhou et al., 2022), MAE (He et al., 2022), EVA (Fang et al., 2023; 2024)) face challenges due to the inherent sparsity of edge maps, which lack the rich semantic information necessary for effective training. For instance, with the MAE algorithm, since most regions of an edge map are white, the task effectively becomes reconstructing original white patches from masked white patches—an inherently difficult learning process. Moreover, because of the high degree of redundancy (Haghighi et al., 2021) and noise (Mahajan et al., 2018) of unlabeled data, and the requires more computational resources and time of supervised pre-training (Chen et al., 2020; Tang et al., 2022), supervised pre-training, with its explicit learning objectives, outperforms unsupervised pre-training (Li et al., 2024).

Based on these considerations, we propose a supervised pre-training scheme tailor-designed for the unsupervised patent retrieval task. During inference, we extract edge maps from both product and patent images and employ a feature encoder to encode these images. We determine the similarity between products and patents by calculating the cosine similarity of the encoded features, thus obtaining the retrieval results. To develop an effective edge map encoder, we adopt a simple yet effective supervised training method: we first extract edge map features from images, then use the corresponding classification results of these edge maps as supervision to train the encoder (He et al., 2016). This approach yields a feature encoder capable of encoding both product and patent images.

Given the current lack of a dataset for image-based patent-product retrieval, we have also collected a large-scale Product-Patent Image Retrieval (PPIR) dataset, which enables testing and unsupervised pre-training. Specifically, we contribute the following resources: **1. PPIR-testing**: a test set with 240 product-patent pairs; **2. PPIR-patent**: a patent retrieval pool with 16,850 patents. **3. PPIR-unlabeled**: a dataset for unsupervised training, containing a total of 3,799,695 images of products and patents. We believe that this dataset will aid the research community in advancing intellectual property rights protection.

## 2 RELATED WORK

For patent analysis, covering tasks such as classification, retrieval, and semantic understanding (Krestel et al., 2021; Shomee et al., 2024). Accurate patent classification (Lu et al., 2020; Yücesoy Kahraman et al., 2023; Li et al., 2023; Bekamiri et al., 2024; Shomee et al., 2024) enables better organization and accessibility of patent information. The potential of AI in automating patent classification has been investigated by several researchers. For example, Kamateri et al. (2024) explored whether AI can effectively solve the patent classification problem, analyzing various machine learning models and their ability to handle the complexity of patent data.

### 2.1 PATENT RETRIEVAL METHODS

AI techniques have been leveraged to improve the accuracy and efficiency of retrieving relevant patents (Shomee et al., 2024). In text-based retrieval, Kang et al. (2020) proposed a patent method prior art search using deep learning language models. Their approach utilizes semantic embeddings to capture the contextual meaning of patent documents, resulting in improved retrieval performance over traditional keyword-based methods. Additionally, Siddharth et al. (2022) enhanced patent retrieval by combining text embeddings with knowledge graph embeddings, demonstrating that integrating external knowledge sources can further refine search results. Lo et al. (2024) proposed to use large language models for patent retrieval.

For image-based retrieval, given that patents often include technical drawings and diagrams, image-based retrieval has emerged as an important area of study. Kravets et al. (2017) focused on improving the quality of convolutional neural network (CNN) (LeCun et al., 1995) training datasets for patent image retrieval, emphasizing the importance of dataset quality in model performance. Building on this, Higuchi et al. (2023) introduced methods using cross-entropy-based metric learning to enhance the accuracy of patent image retrieval systems. In another study, Higuchi & Yanai (2023) proposed transformer-based deep metric learning for patent image retrieval, showcasing the effectiveness of transformer architectures in this domain. Although some works use sketches to retrieve images (Parmar et al., 2024; Koley et al., 2024b;a), sketches and patent images are still different, as patent images usually contain additional noise (e.g., numbers and reference lines), and the lines are thinner than those in sketches. In this work, we aim to address the domain gap in unsupervised retrieval between product and patent images. It is worth noting that, unlike the unsupervised domain adaptation setting (Ganin & Lempitsky, 2015; Pinheiro, 2018; Xu et al., 2023; Gong et al., 2024), where labeled source domain data are available, in our setting we cannot obtain any labeled patent-product image pairs due to the labor-intensive nature of labeling.

### 2.2 PATENT DATASETS AND BENCHMARKS

The availability of large-scale, well-structured datasets is essential for advancing AI research in patent analysis. Risch et al. (2020) presented *PatentMatch*, a dataset specifically created for matching patent claims with prior art, facilitating studies in patent infringement detection and novelty assessment. For image-focused research, Kucer et al. (2022) developed *DeepPatent*, a large-scale benchmarking corpus aimed at patent drawing recognition and retrieval. Ajayi et al. (2023) extended this work with *DeepPatent2*, focusing on technical drawing understanding. Suzgun et al. (2024) introduced the Harvard USPTO Patent Dataset, a comprehensive corpus of patent applications designed to support diverse AI research tasks. However, previous methods mainly focus on image retrieval between patents, with an absence of research on the product-patent retrieval task. Therefore, in this work, we provide a dataset for image retrieval between products and patents.

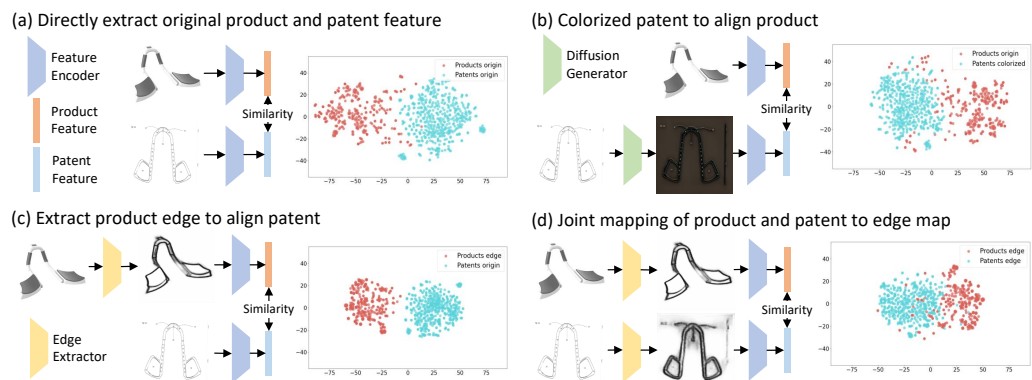

Figure 1: Our method effectively mitigates the domain discrepancy between product and patent images. Products and patents are represented as points in two different colors. The figure contains four subplots, showing the visualization of dimension-reduced features extracted by ResNet50: (a) *Baseline*: original product images and original patent images; (b) *Product2patent*: product edge maps and original patent images; (c) *Patent2product*: original product images and colorized patent images; (d) *JointMap*: both product images and patent images processed using edge extraction. TSNE (Maaten & Hinton, 2008) is used to visualize the feature for matching in 2D. In each subplot, closer interleaving of the two colored points indicates a greater reduction in domain discrepancy.

## 3 METHOD

### 3.1 PPIR: A DATASET FOR PRODUCT-PATENT IMAGE RETRIEVAL

**PPIR Testing Set and Patent Database** Due to the absence of existing datasets containing infringing product and patent image pairs, we constructed a test set to facilitate retrieval from patent images to product images. Specifically, let the product dataset be $D_r$ and the patent dataset be $D_p$. For a product $x_i$ from the product dataset $D_r$, we aim to find its corresponding patent $y_j$ in the patent dataset $D_p$. We represent the labels as the correspondence between infringing products and patent images, forming a set $R = \{(x_i, y_{ij}) \mid i = 1, \ldots, I = 240\}$. In this work, we collected 240 product-patent image pairs to evaluate the unsupervised retrieval of product-patent. Based on the matched product-patent image pairs $R$, we establish the following settings to make our image-based retrieval task practical in real-world situations. Considering that the number of patent images in the database is usually much larger than the number of product images to be retrieved, and the language-based methods are prevailing for the retrieval, we employed a language model Kenton & Toutanova (2019) to match and filter the descriptions of products and patents. For each product image, we obtained the top 2,500 potential corresponding patents. Ultimately, we retained 16,850 items from the patent database for patent-product matching.

**PPIR-unlabeled: A Pre-training Datasets** To evaluate the effectiveness of unsupervised pre-training algorithms, we randomly extracted over 2 million images from the patent database and over 1 million product images from Amazon to pre-train an encoder capable of modeling patent line drawing features. There are 3,799,695 images in total. We first use the edge extractor to extract the edge of these images, then we use the self(un)-supervised learning approaches to pre-train the backbone on the image database.

For the supervised pre-training, we simply adopt the ImageNet1K (Deng et al., 2009) as the training set, which contains 1,281,167 images with 1,000 categories in total. We also extract the edge map of the samples from the ImageNet1K to get ImageNet1K-edge and use the label for classification as supervision.

**Evaluation** Given that our algorithm is designed to assist manual inspection, we assess the retrieval performance by calculating the similarity between patent image features and product image features. In our retrieval task, there is a one vs many situation, which indicates that given that one pair, we have one product image and many patent images. We choose to use the average similarity of all the

pairs to denote the retrieval similarity in this pair. We sort the patents in the matching pool according to their similarity scores and use metrics average rank (AvgRank) and area under the recall curve (AUC). For example, the rank $k$ indicates the average rank of correct matched patents and ranks within the top-$k$ results to evaluate various models.

## 3.2 JOINT FEATURE MAPPING FOR UNSUPERVISED PRODUCT-PATENT IMAGE ALIGNMENT

Retrieving patents for infringing products is complex and time-consuming, heavily relying on patent experts to search through massive patent databases to identify potential infringing patents (Shomee et al., 2024). This reliance makes it challenging to obtain a large-scale paired dataset of product-patent images to train models in a supervised manner. Therefore, this paper focuses on *unsupervised* product-patent image retrieval by leveraging existing feature encoders to obtain high-dimensional features of products and patent images, thereby enabling product retrieval in extensive patent databases.

However, unsupervised methods face a primary challenge: the domain discrepancy between product images and patent images. Specifically, product images are RGB images containing rich color, texture, and other information (Ge et al., 2022), while patent images are line drawings that abstractly represent products (Koley et al., 2023; Voynov et al., 2023). Resolving the domain differences between product and patent images thus becomes our primary task. In this work, we consider three unsupervised methods to achieve feature alignment between patent and product images. These three unsupervised feature alignment approaches and their specific implementations are described as follows.

1. **Converting product images into abstract line drawings for feature alignment with patent images.** We employ an edge feature extractor $E_{\text{edge}}(\cdot)$ (Zhou et al., 2024) to extract edge features from product images $I_r$, converting them into line drawings $I'_r = E_{\text{edge}}(I_r)$. The advantage of this method is its computational efficiency and ease of implementation. However, it heavily depends on the accuracy of the edge feature extraction algorithm. Under varying lighting and texture conditions, the extracted edge maps from product images may exhibit discontinuities, affecting the accuracy of feature alignment. Additionally, we lack an encoder capable of extracting robust features from edge maps to achieve efficient line-drawing representation.

2. **Converting patent images into product images to align features between the two.** Recently, generative models (Ho et al., 2020; Yang et al., 2023; Gong et al., 2024) have gained increasing attention, with many models capable of efficiently generating RGB images based on line drawings (Parmar et al., 2024; Koley et al., 2024b;a). We utilize a generative model $G(\cdot)$ (Parmar et al., 2024) to transform patent images into pseudo-product images, $I'_p = G(I_p)$. The advantage of this method is that the generated images are in RGB format, allowing direct use of feature encoders pre-trained on natural images for feature extraction. However, challenges include the significant computational cost of generative models and the lack of specific domain knowledge, resulting in generated images whose color distribution and texture may differ substantially from natural images.

3. **Simultaneously mapping patent images and product images into a common feature space for unsupervised alignment.** The essence of this method is to use the same image feature extraction $T(\cdot)$ to perform feature transformations on both patent images and product images simultaneously, i.e., $I'_p = T(I_p)$ and $I'_r = T(I_r)$. In this work, we extract edge maps from both patent and product images using the same edge detector $E_{\text{edge}}(\cdot)$ (Zhou et al., 2024), setting $T(\cdot) = E_{\text{edge}}(\cdot)$. After undergoing identical feature mappings, we obtain more similar high-dimensional feature representations. The drawback of this approach is the absence of an efficient edge map encoder capable of extracting discriminative features from the transformed images.

We conducted extensive evaluations of the above three approaches using ResNet-50 (He et al., 2016), pre-trained on ImageNet (Deng et al., 2009), for feature extraction. The results shown in Fig. 1 indicate that simultaneously mapping patent images and product images into a common feature space is beneficial for joint feature matching. However, the lack of an efficient edge map encoder still limits the further performance improvement of this method. In the next section, we explore the supervised pre-training to get a powerful feature encoder on the edge map.

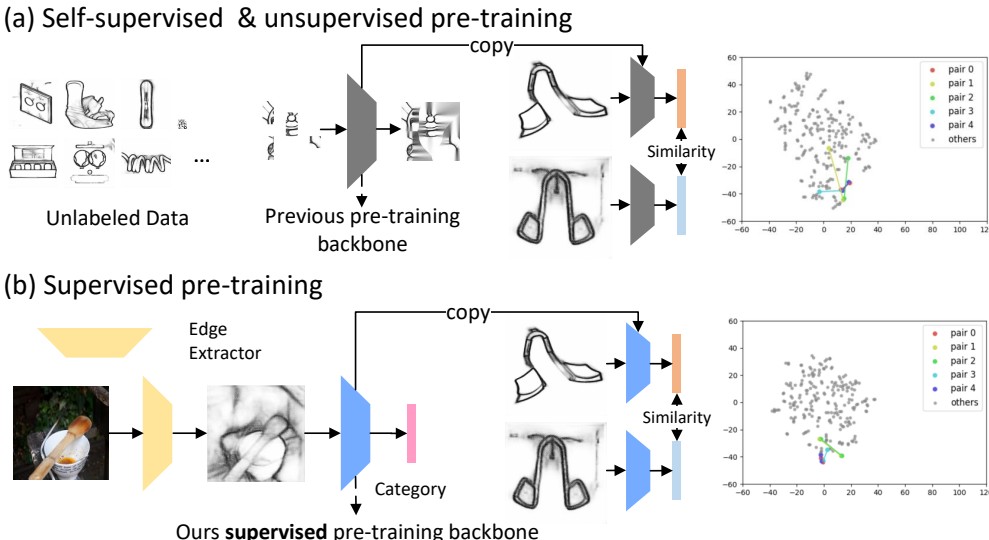

Figure 2: Comparison between self-/unsupervised pretraining and supervised pretraining strategies. In subplot (a), a self-/unsupervised pretraining approach utilizes an ImageNet1k-pretrained backbone applied to the edge maps of product and patent images. Subplot (b) employs our supervised pretraining method using an ImageNet1k-edge supervised backbone. Matched product-patent image pairs are depicted using the same color. The visualization demonstrates that our supervised pretraining method effectively brings matched pairs closer in the feature space, thereby enhancing their alignment.

### 3.3 UNSUPERVISED PRE-TRAINING V.S. SUPERVISED PRE-TRAINING ON PRODUCT-PATENT IMAGE REPRESENTATION LEARNING

Unsupervised pre-training has played a significant role in computer vision recently (He et al., 2022), enabling efficient visual representation encoders without the need for labeled training data. However, recent studies have shown that supervised pre-training can achieve better results than unsupervised pre-training (Li et al., 2024). Focusing on pre-training the edge map encoder, we explore the performance comparison between supervised and unsupervised methods. In this section, all experimental settings are based on the approach described in the previous section, where both product images and patent images are mapped to edge maps. Firstly, we apply a state-of-the-art edge detector $E_{\text{edge}}(\cdot)$ to extract edge maps from the unlabeled images in our large-scale dataset:

$$M_i = E_{\text{edge}}(I_i), \quad \forall I_i \in \mathcal{D}_{\text{unlabeled}}, \tag{1}$$

where $\mathcal{D}_{\text{unlabeled}} = \{I_i\}_{i=1}^N$ is the set of unlabeled images, and $M_i$ represents the edge map of image $I_i$. Subsequently, we utilize unsupervised pre-training methods such as MAE (He et al., 2022), DINO (Caron et al., 2021), and EVA (Fang et al., 2023; 2024) to obtain feature encoders for these edge maps by minimizing the unsupervised loss function $\mathcal{L}_{\text{unsup}}$:

$$\min_F \sum_{i=1}^N \mathcal{L}_{\text{unsup}}\left(F(M_i)\right), \tag{2}$$

where $F(\cdot)$ is the feature encoder we aim to learn. $\mathcal{L}_{\text{unsup}}$ is the unsupervised loss function specific to the pre-training method used. This formalizes the process of applying the edge detector to the unlabeled dataset and then utilizing unsupervised pre-training techniques to learn a feature encoder aligned with downstream edge-based image representations. To construct an unsupervised pre-training task, we use the large-scale unlabeled dataset established in Section 3.1 as training samples. To ensure that this unsupervised pre-training task aligns with downstream edge-based image representations, we first apply a state-of-the-art edge detector $E_{\text{edge}}(\cdot)$ to extract edge maps from

these unlabeled images. Subsequently, we attempt unsupervised pre-training methods such as MAE (He et al., 2022), DINO (Caron et al., 2021), and EVA (Fang et al., 2023; 2024) to obtain feature encoders for edge maps.

Despite the significant progress of unsupervised pre-training in natural image understanding, it faces challenges with patent edge maps due to their sparsity. Many patent images have large blank regions; thus, in masked self-supervised learning schemes, we are likely to mask out regions containing no information, requiring the model to reconstruct these blank areas, which is difficult and inefficient. For contrastive learning methods, they treat different views of the same image as positive samples and other images as negative samples. Due to the sparse features, cropped patches from different images may appear highly similar, further hindering the model's ability to learn correct edge feature representations. To address the shortcomings of unsupervised pre-training, we propose a simple yet effective supervised pre-training method by leveraging existing large-scale labeled datasets such as ImageNet. Specifically, we first convert the ImageNet dataset into edge maps using the edge extractor: $I_{\text{edge}} = E_{\text{edge}}(I_{\text{ImageNet}})$. Based on these edge maps, we train a classifier (feature encoder) $F(\cdot)$ to perform classification according to the original labels of these images. The training objective is to minimize the cross-entropy loss:

$$\mathcal{L} = -\sum_i y_i \log \hat{y}_i, \quad \text{with} \quad \hat{y}_i = \text{softmax}(F(I_{\text{edge},i})), \tag{3}$$

where $y_i$ is the ground truth label, and $I_{\text{edge},i}$ is the edge map of the $i$-th image in ImageNet. The advantage of this approach is that our edge encoder has an explicit learning objective and does not rely on negative samples in contrastive learning or masks in self-supervised learning. This endows our edge encoder with excellent discriminative ability.

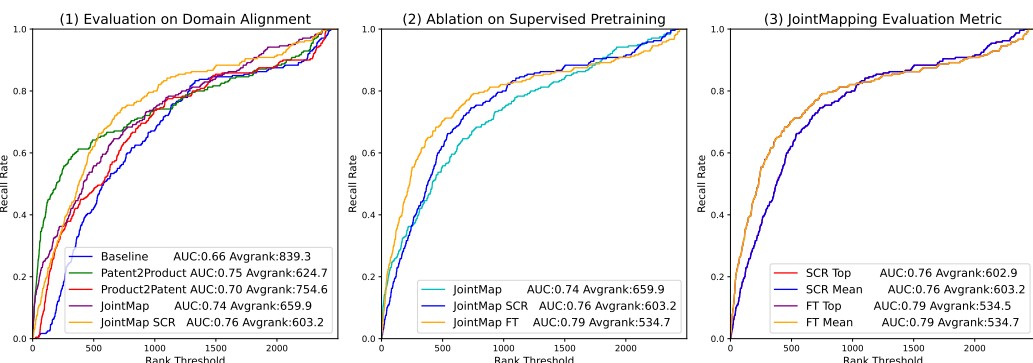

Figure 3: Experiments on mitigating domain gap and ablation study on supervised pre-training. The curve closer to the left upper indicates better results. The performance curves illustrate that approaches with curves closer to the upper-left corner achieve superior results. "SRC" indicates training the backbone from scratch, while "FT" indicates fine-tuning the backbone with the ImageNet pre-trained weight. **(a) Mitigating domain gap:** Analysis on addressing the domain gap between the patent data and image data. The backbone neural network is resnet50. **(b) Ablation on different pre-training methods:** We explore the application of classification-based supervision in unsupervised product-patent retrieval tasks, comparing models trained from scratch on edge maps with those pre-trained on RGB images (i.e., ImageNet pre-training). The results show that our method significantly outperforms retrieval results using natural images. **(c) Ablation on different evaluation methods:** We compute the similarity between the product image and each of the $N$ patent images, obtaining a $1 \times N$ similarity vector. We can either select the maximum value ("Top" in the legend) in this similarity vector to represent the similarity of the product-patent pair or use the average ("Mean" in the legend) of this vector.

## 4 EXPERIMENTS

Our computational setup includes an AMD R9-7950X CPU and 8 NVIDIA V100 GPUs, each with 32GB of VRAM. CUDA Version is 12.4. We spend over 6,000 GPU hours to train the models

mentioned in our work. The detailed hyper-parameters for supervised pre-training and unsupervised pretraining are available in the supplementary material. We are excited to find that our supervised Swin-S network exceeds the best-unsupervised MAE He et al. (2022), with only 1/3 training samples and 1/6 training time.

## 4.1 ANALYSIS ON DOMAIN GAP

In this section, we primarily investigate the unsupervised approaches introduced in Section 3.2 to mitigate the domain gap between product images and patent images. Fig. 3 presents a quantitative analysis of the methods illustrated in Fig. 1 and Fig. 2. It can be observed that both extracting edges from product images and transforming patent images into product images significantly improve the performance of product-patent image retrieval tasks compared to using visual encoders pre-trained on natural images. Furthermore, by using a generative model to convert patent images into product images, their feature spaces are aligned with the domain of natural images, which enables the pre-trained ImageNet encoder to achieve considerable performance. This result is similar to directly mapping both product and patent images to the same feature space using an edge extractor.

However, the aforementioned methods have an efficiency drawback. Considering the massive number of patents that need to be matched in the patent database, even when retrieval is performed via natural language, each inference requires colorizing the images, which is much less efficient than conventional image retrieval. Converting images into edge maps (28 imgs/s) (Zhou et al., 2024) is much more efficient than transforming edge maps into product images (2 imgs/s) (Parmar et al., 2024), and there may also be feature mismatch issues in image colorization. Therefore, we consider whether we can train a supervised edge map feature encoder to achieve better performance. We present this result in Fig. 3-(1). We first extract edges from the ImageNet1k dataset and train a model from scratch without using the pre-trained weights on color images. It can be observed that, compared to image colorization, our method achieves better performance. In summary, these tables demonstrate that both *JointMap* and *Patent2Product* are effective alignment methods. Moreover, by pretraining a backbone network for edge map classification using supervised learning, our method achieves better results.

Considering that ImageNet pre-trained weights are intrinsically available, in Fig. 3-(2) we show a comparison between fine-tuning ImageNet pre-trained backbone networks using edge maps and training from scratch on edge maps. We find that fine-tuning ImageNet pre-trained networks achieves better performance and requires fewer training epochs compared to training from scratch on edge maps. This may be due to the transferability of features learned from natural images to edge maps, as pre-trained models capture low-level features such as edges and textures that are also present in edge maps (Yosinski et al., 2014). In Fig. 3-(3), we explore two different matching methods. In our setting, each product image corresponds to one patent, and each patent contains $N$ images. We compute the similarity between the product image and each of the $N$ patent images, obtaining a $1 \times N$ similarity vector. We can either select the maximum value in this similarity vector to represent the similarity of the product-patent pair or use the average of this vector. We investigate both methods and find that the results show almost no difference (the two curves nearly overlap). Therefore, in our subsequent experiments, we use the average value of the similarity vector to measure the similarity of the product-patent image pair.

## 4.2 ANALYSIS ON SUPERVISED-PRETRAINING

In this section, we explore the content of Section 3.3, comparing supervised pretraining with unsupervised pretraining. Fig. 4-(1) presents the comparison results between our method and existing unsupervised pretraining methods. It can be observed that our method, using only the ImageNet1k dataset (compared to our unsupervised pretraining on PPIR-unlabeled), achieves better performance with only one-third of the training data. Models pre-trained via self/unsupervised methods were assessed to determine their effectiveness. We included models like iBOT (Zhou et al., 2022), MAE (He et al., 2022), DINO (Caron et al., 2021), EVA (Fang et al., 2023), EVA02 Fang et al. (2024), Swin-S (Liu et al., 2021), and Swin-S FT. The performance metrics, specifically Area Under the Curve (AUC) and Average Rank (AvgRank) were analyzed. The Recall Rate for these models was charted against Rank Thresholds ranging from 0 to 2000. MAE exhibited outstanding performance, achieving the highest AUC of 0.79 and an AvgRank of 522.5. It is worth noting that, our method only

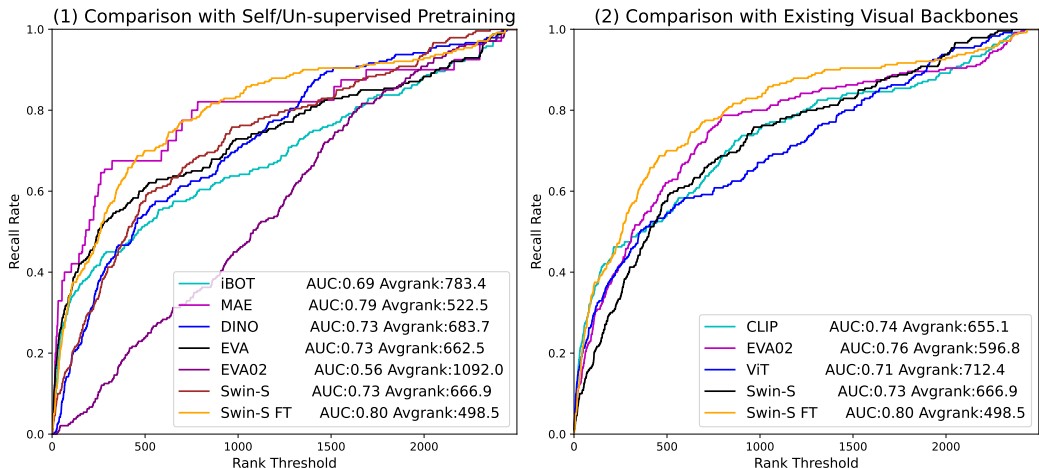

Figure 4: Comparison with the self/un-supervised methods and existing visual backbones. "FT" indicates our supervised fine-tuning. **(1) Self/un-supervised methods:** We compared our method trained on the ImageNet1k-edge and the Comparison with the self(un)-supervised pre-training methods (Zhou et al., 2022; Caron et al., 2021; He et al., 2022; Fang et al., 2023; 2024) methods training on the PPIR-unlabeled. **(2) Existing visual backbones:** We also compared our method with other methods trained on the powerful visual backbones (Liu et al., 2021; Fang et al., 2024). The result shows that our method is extremely powerful.

trained on the ImageNet1K-edge with about 1.3 million samples for 16 hours, while the MAE is trained on the PPIR-unlabeled with 3.8 million samples for about 100 hours. This suggests its effectiveness in feature extraction even without supervised learning. Swin-S FT also showed impressive results with an AUC of 0.80 and an AvgRank of 498.5, indicating that fine-tuning has significantly enhanced its capability.

In Fig. 4-(2), we compare our method with some of the current state-of-the-art vision backbone networks, such as EVA02 (Fang et al., 2024) at $448 \times 448$ resolution, which are trained on multiple large-scale datasets. It can be seen that through our supervised pretraining, using only 1 million classification images at $224 \times 224$ resolution. This subset of the evaluation focused on comparing the aforementioned models against traditional visual backbones like CLIP, EVA02, VIT, and Swin-S. The objective was to evaluate whether recent advancements in self/unsupervised learning could match or surpass the performance of conventional supervised learning models. EVA02 and Swin-S FT reached an AUC of 0.76 and 0.80 respectively, with Swin-S FT maintaining a particularly low AvgRank, which reinforces the effectiveness of its fine-tuning approach. CLIP maintained a robust performance with an AUC of 0.74 and an AvgRank of 655.1, demonstrating its adaptability across various visual tasks.

The findings from this comparison demonstrate that certain unsupervised and fine-tuned models are competitive with, or superior to, traditional supervised models, offering promising alternatives for robust visual recognition systems.

### 4.3 ANALYSIS ON SCALEUP ABILITY

Fig. 6 compares the performance of our method across different network architectures and parameter sizes, including CNN-based (LeCun et al., 1995) ResNets (He et al., 2016) and Transformer-based (Vaswani et al., 2017) SwinNets (Liu et al., 2021). It can be observed that through our proposed edge-based pretraining method, we achieve consistent performance improvements on the patent-product image retrieval task. Specifically, our methods improve the performance by about 3% AUC score on ResNets, and by about 4% AUC score on SwinNets. We also witness a boost in the reduction of average rank.

However, we also find that neither the pre-trained weights nor our edge fine-tuned weights show a direct correlation between performance improvement and parameter size. We consider that the potential reason lies in the fact that, despite extracting edge images, there are inherent differences between the edge maps of patent images and those of natural images. Moreover, our current test set is not sufficiently large, and further expanding the test set is a direction for our future work.

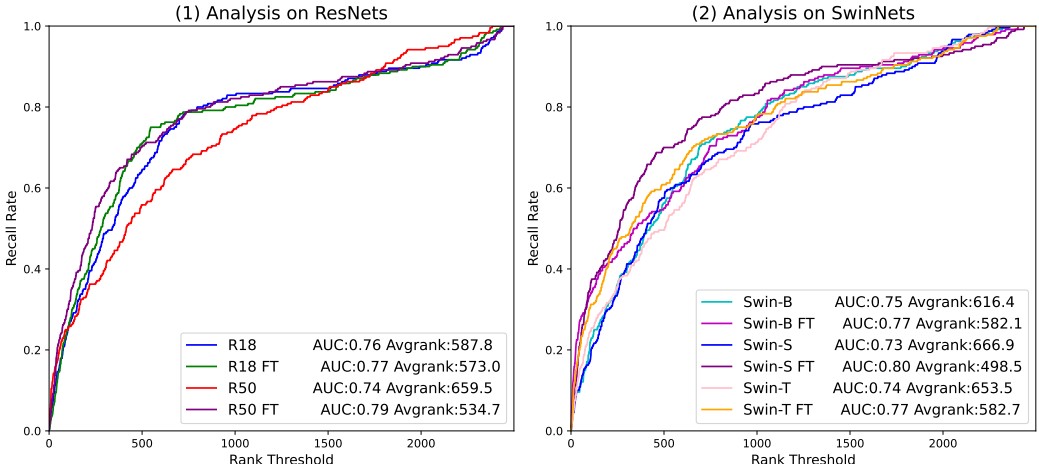

Figure 5: **Scaling up the model:** We trained 5 different backbones: ResNet-18, ResNet-50 (He et al., 2016), Swin-Tiny, Swin-Small, and Swin-Base (Liu et al., 2021) on ImageNet1K-edge. The results indicate that our method can significantly and consistently boost unsupervised patent-product retrieval performance.

## 5 CONCLUSION AND DISCUSSION

In this paper, we explored a novel product-patent image retrieval task and conducted an in-depth investigation into the application of supervised pretraining based on edge maps within this context. We began by constructing a retrieval pair dataset, a patent database for retrieval, and a patent-product database for unsupervised pretraining. The datasets provided in this paper establish an effective method for evaluating both unsupervised and supervised patent retrieval tasks.

Our experimental results demonstrate that mapping both patent images and product images into edge maps or colorizing patent images into product images, are effective strategies for alleviating the domain gap. Furthermore, by simply converting existing classification data into edge maps and setting up a supervised edge classification task, we can enhance the feature extraction performance of our backbone network. Besides, we have make the following findings through our extensive experiments: **1. Mapping patent and product to joint edge representation, or transform the patent image to product is beneficial for alleviate the domain gap. 2. Supervised pre-training with classification label is much more efficient than unsupervised pre-training on the edge map.** In the future, we will continue to scale up our testing set and focus on addressing the domain gap between the patent edge map and the product edge map.

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

# A  APPENDIX ON TRAINING DETAILS

Table 1: Hyper-parameters of Supervised training on ResNet

| Hyper-parameters | Value |
| --- | --- |
| --model | ResNet-18 / 50 / 101 |
| --dataset | ImageNet1K-edge |
| --batch_size | 4096 / 2048 / 1024 |
| --optimizer | AdamW |
| --learning rate | 0.002 |
| --opt_betas | (0.9, 0.95) |
| --warmup_epochs | 5 |
| --epochs | 100 |
| --weight_decay | 0.001 |
| --training time | 5.3h / 9.5h / 18.7h |

Table 2: Hyper-parameters of Supervised training on SwinTransformer

| Hyper-parameters | Value |
| --- | --- |
| --model | Swin-T / S / B |
| --dataset | ImageNet1K-edge |
| --batch_size | 2048 / 1024 / 512 |
| --optimizer | AdamW |
| --learning rate | 0.002 |
| --opt_betas | (0.9, 0.95) |
| --warmup_epochs | 5 |
| --epochs | 100 |
| --weight_decay | 0.001 |
| --training time | 7.9h / 15.7h / 21.6h |

Table 3: Hyper-parameters of Supervised training on VisionTransformer

| Hyper-parameters | Value |
| --- | --- |
| --model | ViT-B-16 / L-16 |
| --dataset | ImageNet1K-edge |
| --batch_size | 512 / 256 |
| --optimizer | AdamW |
| --learning rate | 0.002 |
| --opt_betas | (0.9, 0.95) |
| --warmup_epochs | 5 |
| --epochs | 100 |
| --weight_decay | 0.001 |
| --training time | 19.8h / 34.7h |

Table 4: Hyper-parameters of Self-Supervised Contrastive Learning

| Hyper-parameters | Value |
|---|---|
| --method | DINO / iBOT |
| --model | ViT-L-16 |
| --dataset | PPIR-unlabeled |
| --batch_size | 256 |
| --optimizer | AdamW |
| --learning rate | 0.002 |
| --opt_betas | (0.9, 0.95) |
| --warmup_epochs | 40 |
| --epochs | 100 |
| --training time | 97.4h / 98.8h |

Table 5: Hyper-parameters of Masked Image Model

| Hyper-parameters | Value |
|---|---|
| --method | MAE / EVA / EVA-02 |
| --model | ViT-L-16 |
| --dataset | PPIR-unlabeled |
| --batch_size | 256 |
| --optimizer | AdamW |
| --learning rate | 0.002 |
| --opt_betas | (0.9, 0.95) |
| --warmup_epochs | 40 |
| --epochs | 100 |
| --training time | 95.3h / 96.5h / 99.2h |

# B  VISUALIZATION RESULTS

(a) Visualizations on generated product image based patent image with diffusion model

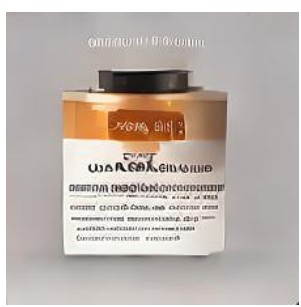 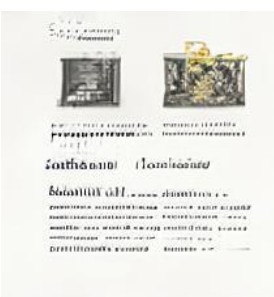 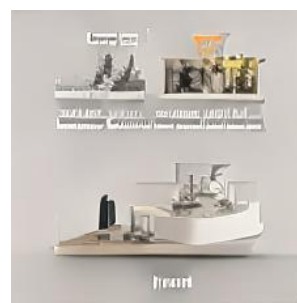

(b) Visualizations on extracting edge from ImageNet1k

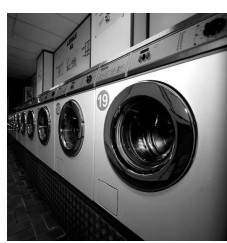 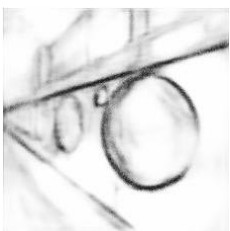 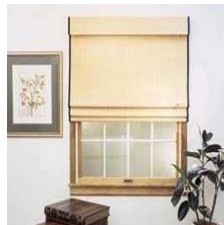 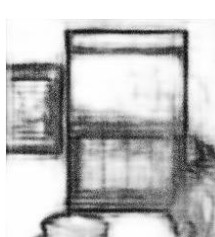

Figure 6: This figure provides some visualization results on the diffusion generated samples and edge map from ImageNet1k.

