# OpenReview forum: "Supervised Pre-training for Unsupervised Product-Patent Image Retrieval"
_ICLR.cc/2025/Conference — ICLR 2025 Conference Withdrawn Submission_

### Official Review · Reviewer_fwZP · 2024-10-28

**Soundness:** 2
**Presentation:** 2
**Contribution:** 2
**Rating:** 3
**Confidence:** 5

**Summary:**

This work aims to alleviate the domain discrepancies between patents and products. A supervised pretrained method is designed for patent image retrieval.

**Strengths:**

The authors construct a dataset for patent-product image retrieval, which includes product-patent pairs and unlabeled data.

**Weaknesses:**

I dislike the motivation behind this research. As far as I know, there is a significant domain gap between product images and patent images, making it meaningless to align the domain gap between them. Patent images are line drawings derived from binarized sketches, whereas product images are rich in color and textual information. The imaging methods corresponding to patent images and product images differ significantly, resulting in vastly different content forms they represent.

The author does not compare their method with the state-of-the-art (SOTA) approaches (e.g., [1]) for patent image retrieval, and these methods are also not reviewed in the related work section.  The work DeepPatent has been cited 25 times, and the proposed method should be compared with DeepPatent and subsequent representative methods.

Constructing the dataset is an innovation of this paper, yet the author does not provide detailed information about the dataset. How are the paired data of (patent image, product image) constructed? And how can we ensure their semantic identity or identity at the instance level?

The method presented in this paper lacks innovation, as it merely applies pretraining techniques to patent images.

The title is confusing. Both words Supervised  and Unsupervised are appeared in the title.

[1] DeepPatent: Large scale patent drawing recognition and retrieval. Michal Kucer , Diane Oyen , Juan Castorena , Jian Wu

**Questions:**

N/A

---

### Official Review · Reviewer_S1aL · 2024-10-30

**Soundness:** 2
**Presentation:** 3
**Contribution:** 2
**Rating:** 5
**Confidence:** 3

**Summary:**

The paper addresses the challenge of ensuring new products do not infringe on existing patents through a product-patent retrieval task. It targets the domain discrepancies between patent images (typically line drawings) and product images (photographs). It studies unsupervised and supervised methods to align these domains and improves a variety of models with enhanced feature extraction. The authors create a large dataset of over 3.7 million images to support their model training and 240 product-patent image pairs for testing. While their experiments yield promising results, the authors acknowledge limitations regarding the test set size and the relationship between model size and performance.

**Strengths:**

The work addresses a practical problem in patent protection, which is in particular critical as new products emerge in the market.

The work makes a contribution with the creation of a large dataset for training and a test dataset of images. The posed questions provide a focused framework for exploring the challenges of domain adaptation in patent-product image retrieval.

**Weaknesses:**

The paper lacks robust validations to demonstrate that the proposed approach is superior to other alternative methods. For example, in the context of cartoon and photo retrieval or sketch and image retrieval [1,2,3,4], various methods have been discussed and developed. Although there are differences between patent images and sketches or cartoons, these methods still serve as potential strong baselines. Additionally, some of these methods leverage generative models to address the lack of training data and bridge the domain gap between different types of visual content [5][6], providing alternative solutions to the challenges presented in this work. The lack of such comparisons weakens the conclusions drawn in this paper.

The work appears to be quite specific to the domain of patent protection, which may limit its appeal to the broader research community. The authors could explore potential applications beyond patents. For example, their approach for aligning line drawings with photographs might be applicable in areas such as technical diagram interpretation or historical document analysis. Highlighting these broader applications could help demonstrate the wider impact of this work and attract greater interest from diverse research fields.

[1] Deep Sketch Hashing: Fast Free-hand Sketch-Based Image Retrieval;

[2] A Systematic Literature Review of Deep Learning Approaches for Sketch-Based Image Retrieval: Datasets, Metrics, and Future Directions;

[3] Recent Advances in Sketch Based Image Retrieval: A Survey;

[4] Cartoon Based Image Retrieval : An Indexing Approach;

[5] SketchyGAN: Towards Diverse and Realistic Sketch to Image Synthesis;

[6] CartoonGAN: Generative Adversarial Networks for Photo Cartoonization

**Questions:**

The paper could benefit from a more comprehensive exploration of related work in similar domains, as mentioned in the weaknesses. I expect deeper comparisons that go beyond surface-level discussions. This may help further establish its contributions to the field.

---

### Official Review · Reviewer_T4CT · 2024-10-30

**Soundness:** 2
**Presentation:** 2
**Contribution:** 2
**Rating:** 3
**Confidence:** 4

**Summary:**

This paper attempts to tackle product-patent retrieval task in the image modality. Due to the blank in previous research, the authors constructed a retrieval pair dataset, a patent database for retrieval, and a patent-product database for unsupervised pretraining. At the same time, the authors tried to combine the edge extraction module to project specific patent and product images into the same feature space for retrieval. In addition, the authors also proposed a pre-training scheme. The experiments have proved that the method has certain advancedness.

**Strengths:**

1. The addressed task in this paper is well-motivated and seems to be reproducible.
2. This paper is easy to follow.
3. The paper focuses on solving product-patent retrieval task that is of application importance, and the results obtained seem to be quite good.

**Weaknesses:**

1. What I am most concerned about is the motivation of patent-product retrieval in image modality. Compared with text, can it retrieve the corresponding product more accurately? Authods could conduct experiments comparing their image-based method to a text-based baseline to verify their motivation. Additionally, could using multimodal collaboration provide a more direct solution for current patent-product issues, rather than relying only on image matching? Finally, in my opinion, the method is based on the assumption that patent images are all line drawings and hand-drawn drawings, and uses edge extraction to specifically compensate for cross-domain offsets. In practice, for example, 3D rendered images and so on will also appear in patents. Is a more general domain adaptation module preferable?
2. Lack of theoretical innovation. The paper seems to be mature in pre-training methods, and lacks detailed descriptions in Sec. 3.3.
3. The review of related work is not comprehensive. The related work of image modality pre-training is not reflected in Sec. 2, which is extremely relevant in the main contribution of this paper.
4. More analysis and visualization of datasets are necessary, such as sample illustrations, systematic statistics, and analysis.  This helps readers understand the data set more clearly.
5. Current experiments seem to be scarce. More rich and in-depth visualization and analysis experiments are needed to prove the quality of the data set and the excellence of the pre-training mechanism.

**Questions:**

A more solid motivation discussion and detailed analysis can help me better understand and appreciate this paper.

---

### Official Review · Reviewer_U6jV · 2024-11-04

**Soundness:** 2
**Presentation:** 1
**Contribution:** 2
**Rating:** 3
**Confidence:** 5

**Summary:**

The paper introduces a simple supervised pre-training method for product-patent image retrieval. The authors focus on the gap between patent images and product images with domain alignment. Specifically, the authors introduced a Product-Patent Image Retrieval dataset, and also observed and evaluated different strategies to project patent and product images into feature spaces. Moreover, the authors proposed to train a classifier on edge maps to improve performance.

**Strengths:**

1. The paper addresses a critical gap in intellectual property protection by focusing on image-based product-patent retrieval. It is an unexplored but important domain.
2. The introduction of PRIR dataset contributes resources for future research in the domain.
3. The authors illustrate practical solutions for product-patent retrieval, which used supervised pre-training on edge maps for domain alignment .

**Weaknesses:**

1. The writing is generally clear, but there are typos, redundant sentences, and many unexplained terms. These make the paper hard to follow. See questions.
2. It is not clear how to build the PRIR dataset, like how to get Amazon product images and use what views of patent images.
3. The authors proposed supervised pre-training on edge maps on ImageNet1K, but ImageNet1K contains many natural images which also can bring domain gaps.
4. The proposed method heavily rely on the edge extractor, so could different extractors affect the performance? How to choice the extractor?

**Questions:**

1. Writing related questions: line 307-313 are almost same as line 323-326; line 323: what are contrastive learning methods here? why mentioned different views here? line 362: "SRC", but in legends it said "SCR"; line 402: these tables, but I didn't see any tables.
2. Patent images contain many views, so did authors filter out other views in the dataset or consider them in the evaluation?
3. In Eq2, what is the unsupervised loss function here?
4. Since patent and product data are domain-specific, would authors consider finetuning on product dataset instead of ImageNet1K?

---

### Note · Authors · 2024-11-13

I have read and agree with the venue's withdrawal policy on behalf of myself and my co-authors.